# Dietary Protein Intake in Older Adults from Ethnic Minorities in the Netherlands, a Mixed Methods Approach

**DOI:** 10.3390/nu13010184

**Published:** 2021-01-09

**Authors:** Elvera Overdevest, Berber G. Dorhout, Mary Nicolaou, Irene G. M. van Valkengoed, Annemien Haveman-Nies, Halime Oztürk, Lisette C. P. G. M. de Groot, Michael Tieland, Peter J. M. Weijs

**Affiliations:** 1Center of Expertise Urban Vitality, Amsterdam University of Applied Sciences, 1067 SM Amsterdam, The Netherlands; h.ozturk@hva.nl (H.O.); m.tieland@hva.nl (M.T.); p.j.m.weijs@hva.nl (P.J.M.W.); 2Division of Human Nutrition and Health, Wageningen University & Research, 6700 EW Wageningen, The Netherlands; berber.dorhout@wur.nl (B.G.D.); lisette.degroot@wur.nl (L.C.P.G.M.d.G.); 3Amsterdam UMC, Department of Public and Occupational Health, Amsterdam Public Health Research Institute, University of Amsterdam, Meibergdreef 9, 1012 WX Amsterdam, The Netherlands; m.nicolaou@amc.uva.nl (M.N.); i.g.vanvalkengoed@amc.uva.nl (I.G.M.v.V.); 4Chair Group Consumption and Healthy Lifestyles, Wageningen University & Research, 6700 EW Wageningen, The Netherlands; annemien.haveman@wur.nl; 5Department of Nutrition and Dietetics, Amsterdam University Medical Centers, VU University, 1081 HV Amsterdam, The Netherlands

**Keywords:** protein intake, ethnic minorities, older adults, sarcopenia, mixed methods

## Abstract

Optimizing protein intake is a novel strategy to prevent age associated loss of muscle mass and strength in older adults. Such a strategy is still missing for older adults from ethnic minority populations. Protein intake in these populations is expected to be different in comparison to the majority of the population due to several socio-cultural factors. Therefore, the present study examined the dietary protein intake and underlying behavioral and environmental factors affecting protein intake among older adults from ethnic minorities in the Netherlands. We analyzed frequency questionnaire (FFQ) data from the Healthy Life in an Urban Setting (HELIUS) cohort using ANCOVA to describe dietary protein intake in older adults from ethnic minorities in the Netherlands (N = 1415, aged >55 years, African Surinamese, South Asian Surinamese, Moroccan, and Turkish). Additionally, we performed focus groups among older adults from the same ethnic minority populations (N = 69) to discover behavioral and environmental factors affecting protein intake; 40–60% of the subjects did not reach minimal dietary protein recommendations needed to maintain muscle mass (1.0 g/kg bodyweight per day (BW/day)), except for Turkish men (where it was 91%). The major sources of protein originated from animal products and were ethnic specific. Participants in the focus groups showed little knowledge and awareness about protein and its role in aging. The amount of dietary protein and irregular eating patterns seemed to be the major concern in these populations. Optimizing protein intake in these groups requires a culturally sensitive approach, which accounts for specific protein product types and sociocultural factors.

## 1. Introduction

In the Netherlands, the number of non-western older adults is rising rapidly, and is expected to grow even further, from 70,000 in 2010 to more than 500,000 in 2050 [1]. In many high-income countries, ethnic minority groups show a worse disease risk profile from a younger age and have lower life expectancies compared to the majority population [2]. For instance, in the Netherlands, chronic diseases and physical limitations appear, on average, ten years earlier compared to the Dutch majority population [3,4,5,6,7]. A major risk factor for developing chronic diseases and physical limitations, is sarcopenia; the age-associated loss of skeletal muscle mass, muscle strength, and physical performance [8]. A recent paper described higher probable sarcopenia rates among older ethnic minority groups in the Netherlands (OR: 0.96, 95% CI: 0.92–0.99) indicating a higher risk of developing chronic diseases compared to the Dutch majority population [9].

The progression of sarcopenia is caused by multiple lifestyle factors, including inadequate dietary protein intake [10]. To minimize the risk of developing sarcopenia, nutritional strategies are needed to help preserve or improve skeletal muscle mass, strength and function in older adults from ethnic minority populations [8,11]. It is yet unknown on which aspects of protein intake interventions should focus on in older adults from ethnic minority populations. Several Randomized Controlled Trials (RCTs) show the beneficial effects of protein and amino acid supplementation on muscle mass, muscle strength, and/or physical functioning [12,13,14,15,16]. Based on these, recommendations on protein intake for older adults have been formulated. For instance, the European Society for Clinical Nutrition and Metabolism (ESPEN) expert groups recommend that healthy older adults consume at least 1.0–1.2 g of protein per kilogram bodyweight per day (g/kg BW/d) to maintain muscle mass [12,17], which is in contrast to the lower recommended dietary allowance of 0.8 g/kg BW/d for healthy adults [18]. In addition to the total amount of protein intake, the source and timing of protein is of importance to optimize the effect of the daily protein intake on prevention of sarcopenia in older populations. Animal proteins, especially those from dairy, are higher quality proteins and seem to better support muscle protein synthesis than plant proteins [19]. Furthermore, protein distribution or timing may impact muscle protein synthesis and muscle mass gain. Increasing the protein intake at breakfast, lunch, and prior to sleep represent effective strategies to stimulate muscle protein synthesis and muscle mass growth [20,21].

Besides, information to determine how interventions may be improved to account for the specific needs of minority populations is scarce. Social and cultural factors differ between ethnic groups and affect daily health behaviors of individuals [22]. Previous studies have highlighted the role of culture in dietary habits [23,24,25]. These studies describe behavioral factors that determine food patterns in general, but more insight is needed in factors that affect, or may explain, dietary protein intake. We expect that older adults from ethnic minorities lack awareness and knowledge about the importance of dietary protein intake while aging due to their generally lower educational and health literacy levels [26]. We also expect that the choice for specific protein sources strongly depend on sociocultural determinants within the different older ethnic groups. Clear insights in these factors in older adults from ethnic minorities are still lacking.

We carried out a mixed methods study to examine quantitatively the dietary protein intake, including protein sources, and explore qualitatively its underlying personal and environmental behavioral determinants among older adults from ethnic minority groups in the Netherlands. The insights from our results provides a window of opportunity to develop an intervention aimed at optimizing protein intake in older ethnic minority populations in a real-life setting.

## 2. Materials and Methods

### 2.1. Quantitative Research

For the cross-sectional analysis, baseline data were used from the Healthy Life in an Urban Setting (HELIUS) study; a cohort study that aims to gain insight in the causes of the unequal burden of disease across ethnic groups [27]. The baseline data were collected between 2011 and 2015, among nearly 25,000 participants aged 18–70, recruited by the municipality of Amsterdam, the Netherlands.

#### 2.1.1. Study Population

A subsample of the HELIUS study population completed ethnic-specific food frequency questionnaires (FFQs; N = 5276) [28]. We selected participants older than age 55 in the current study (N = 1430). Subsequently, participants who missed values for dietary intake or showed extreme values for total energy intake (for men <800 kcal per day or >4000 kcal per day; for women <500 kcal per day or >3500 kcal per day) were excluded. This resulted in a subsample of 1415 subjects who were included for further analyses. The population characteristics are described in the results section and can be found in Table 1.

#### 2.1.2. Measures

Country of birth of participants and their parents were used to determine ethnicity for the HELIUS study [29]. The definitions of each ethnic group and criteria for inclusion to one of the ethnic groups was described elsewhere [27].

Body weight and height were measured in duplicate based on standardized protocols. The mean was used for analysis. A third measure was performed if the two measurements differed more than 0.5 kg (weight) or 0.5 cm (height). In that case, the two measurements closest together were averaged. Body mass index (BMI) was calculated as weight (kg)/height (m)^2^.

Dietary intake was measured using ethnic specific semi-quantitative FFQs, specifically designed for the HELIUS study [28]. Data from the Food Composition Table 2011 were used to construct a nutrient database for each FFQ (Dutch Food Composition Table 2011). For ethnic specific foods, the database was expanded with international data and new chemical analyses. Participants reported the frequency and portion size of 238 food items eaten in the previous four weeks. From the FFQ total energy intake, total dietary protein intake and animal and plant derived protein intake were determined. Energy intake is formulated in kcal/day. Protein intake is presented as g/day and g/kg BW/day as well as a percentage of total energy intake (EN%). Moreover, the protein intake on product level was assessed.

#### 2.1.3. Statistical Analyses

Characteristics were stratified by sex and ethnic background. Age, BMI, total dietary protein intake, and animal and plant derived protein intake are presented as means +/− standard deviations of the mean. To compare the protein intake between ethnic minority groups and the Dutch, which was considered as the reference population, an ANCOVA analysis was performed, controlling for age, chronic diseases, physical activity, smoking, alcohol use, and energy intake.

Proportions of the ethnic populations reaching the recommended daily allowance (RDA) (0.8 g protein/kg BW/day) and the recommended 1.0–1.2 g protein/kg BW/day stated by the ESPEN discussion group [17] were presented and compared for the different ethnic minority populations.

To gain insight in the food sources that contribute most to daily protein intake among the different ethnic groups, we summed the absolute protein intake per product in gram/day for the whole group.

A *p*-value < 0.05 was considered statistically significant. The quantitative analyses were performed using SPSS Statistics (version 24, IBM, Armonk, NY, USA).

### 2.2. Qualitative Research

#### 2.2.1. Recruitment, Study Population and Moderation of Focus Groups

To gain in-depth insights into behavioral factors affecting protein intake, we conducted focus group discussions. Older adults with Surinamese African, Surinamese South Asian, Turkish, and Moroccan origin were consulted in separate focus group discussions. These ethnic minority groups are the largest non-western minority groups living in the Netherlands. Subjects were recruited at community centers in Amsterdam and the region of Apeldoorn. After written informed consent, participants were assigned to one of the focus groups. The focus group discussions with both Surinamese groups were conducted in Dutch and discussions with Turkish and Moroccan older adults were guided by trained native Turkish and native Moroccan moderators. This enabled the focus group participants to speak in their native language.

#### 2.2.2. Discussion Guide

The qualitative study focused on behavioral determinants affecting protein intake, both personal and environmental. First, we explored personal determinants, such as current knowledge and awareness, about protein and its role in healthy aging. Second, we explored environmental determinants that explained dietary (protein) behavior and habits. Participants were asked to come up with associations and explanations on the topic of protein and aging (i), followed by a short informative presentation by the researchers filling in the knowledge gap on this topic. Second, questions were asked about dietary habits, food patterns, and sociocultural factors affecting dietary (protein) intake (ii).

#### 2.2.3. Qualitative Analyses

The audio recordings of all focus group discussions were transcribed verbatim and translated by bilingual members of the research team into Dutch within the software program MaxQDA Analytics Pro 18 (version 2018). A random sample of the translations were checked by an independent native speaker. A framework approach based on the discussion guide was used as base for the coding scheme to analyze the transcripts. Two researchers agreed upon a coding scheme and assigned codes independently to the same transcripts. The coding scheme distinguished behavioral determinants affecting dietary protein intake and timing, both related to the person (e.g., knowledge, awareness, attitude), and environment (e.g., social norms and cultural factors). To ensure intercoder reliability, the two researchers discussed coding differences until consensus was reached. Then codes were rearranged and clustered by axial coding according to the research questions to be answered. After restructuring the codes, the results section was formulated by re-reading segments and linking codes and coded segments from the different transcripts through an iterative process.

## 3. Results

### 3.1. Quantitative Study

#### 3.1.1. Subject Characteristics

The mean age of all ethnic groups ranged from 58.5 to 61.7 years. In men, the lowest BMI was found in Dutch (26.3 ± 3.7 kg/m^2^) and the highest in Moroccan (28.2 ± 4.1 kg/m^2^). In women, the lowest BMI was also found in Dutch (25.9 ± 4.5 kg/m^2^) and the highest in Turkish (32.9 ± 6.1 kg/m^2^). South Asian men and African Surinamese women reported on average the lowest energy intakes (2130 ± 751 kcal/day and 1748 ± 634 kcal/day, respectively). In Turkish participants, both men and women reported, on average, the highest energy intake (2331 ± 648 kcal/day in men and 2122 ± 699 kcal/day in women) (Table 1).

#### 3.1.2. Amount of Protein

Mean relative protein intake ranged from 1.04 ± 0.33 g/kg BW/day in Dutch women to 1.38 ± 0.38 g/kg BW/day in Turkish men, with a significant difference in South Asian Surinamese and Turkish men, in comparison to the Dutch reference population (*p* = 0.024 and *p* = 0.001 respectively; Table 1). Adjusting for chronic disease (self-reported history (or presence) of diabetes, myocardial infarction, or stroke), smoking status, alcohol consumption, or physical activity did not change these results (data not shown). Furthermore, the majority of the study population reached the RDA of 0.8 g with the highest percentage in Turkish men (97%) and the lowest percentage in African Surinamese women (59%). Between 49% and 60% of people in all older populations reached the minimal ESPEN protein recommendation of 1.0 g/kg BW/day, except the Turkish men, of which 91% met this guideline (Table 1).

#### 3.1.3. Protein Source

Next to total protein intake, Table 1 shows the protein intake in g/kg BW/day in the different older ethnic groups by source and stratified by gender. Mean animal derived protein intake was lowest in both Dutch men and women (0.63 ± 0.22 g/kg BW/day and 0.61 ± 0.24 g/kg BW/day, respectively) and the highest in both Turkish men and women (0.86 ± 0.36 g/kg BW/day and 0.68 ± 0.35 g/kg BW/day respectively), with a significant difference in Turkish men compared to Dutch men (*p* = 0.001). Mean plant protein ranged from 0.38 ± 0.18 g/kg BW/day in African Surinamese women up to 0.53 ± 0.28 g/kg BW/day in Moroccan men and differed significantly in South Asian Surinamese men (*p* = 0.024), Turkish men (*p* = 0.010), South Asian Surinamese women (*p* = 0.001), and African Surinamese women (*p* = 0.033) compared to the Dutch reference population. Adjusting for chronic disease (self-reported history (or presence) of diabetes, myocardial infarction, or stroke), smoking status, alcohol consumption of physical activity did not change these results (data not shown).

#### 3.1.4. Protein Product Sources

Ethnic differences were observed concerning the contribution of specific food sources to total daily protein intake and were different from the Dutch majority population, where Dutch cheese and milk contributed most to daily protein intake. In general, the most important sources of protein are ethnic specific. Despite the fact that bread is not considered to be a protein rich source, different types of bread made a large contribution to daily protein intake in all ethnic groups. In both Surinamese groups, chicken and dried fish were typical protein rich products to consume. Turkish older adults derived most of their dietary protein from different types of bread and different types of meat (veal, minced beef, lamb). The most important protein rich products consumed by Moroccan participants were comparable to the Turkish group, except for some animal derived products such as milk, chicken, and crustaceans (Appendix A).

### 3.2. Qualitative Study

#### 3.2.1. Subject Characteristics

In total, 11 focus group discussions were held among older adults from different ethnic groups (Table 2). Most participants were women, except for the Moroccan group where most participants were men.

#### 3.2.2. Qualitative Exploration of Behavioral Determinants of Protein Intake

(1) Personal determinants affecting protein intake: during the focus group discussions, all ethnic groups seem to lack knowledge and awareness of the role of protein in healthy aging. Participants made only a few associations to the term “protein”, when the moderator asked for it; often associations were incorrect and participants were not able to substantiate their answers. This is illustrated by questions asked by participants about the role of protein in a healthy diet and participants showing strong beliefs about the role of protein and other nutrients in losing weight. In other words, protein was mostly related to losing weight rather than maintaining muscle mass and function.

“I don’t know why, but I do know that I should eat more protein and avoid carbohydrates”.

“Don’t you lose weight by eating protein? Please explain to us”!

Indeed, most participants also indicated not knowing which products to choose and what is needed to consume an adequate amount of protein. However, participants were motivated to learn more about the role of protein for healthy aging and daily functioning. This is based on the curiosity participants showed by asking further questions on how to improve their protein intake, especially in Turkish and South Asian Surinamese groups.

“I want to know which products I should take, there are so many, it is confusing”!

“I have a question; I am a vegetarian, what food can I eat to reach a sufficient amount of protein”?

The participants showed their willingness to adopt new dietary habits towards a more protein rich diet, but felt unable to do so. Although participants in each ethnic group showed a positive attitude towards eating an adequate amount of protein, they preferred to receive more information about the role of protein and exact protein sources first. An expected barrier related to optimizing dietary habits, and, thus, improving protein intake, mentioned by participants, is the irregularity of daily meal consumption. Often meals are skipped, postponed, or even combined with other meals. Reasons that were mentioned are a lack of appetite, having an irregular daily schedule, waking up late, suffering from pain or forget to eat.

“I preferably stay in bed, rather than consuming my breakfast. No, I don’t eat breakfasts”.

(2) Environmental determinants affecting protein intake: although their positive attitudes to optimize protein intake, the respondents showed a lack of confidence in their ability to change dietary habits towards optimizing protein intake due to several environmental factors that affect dietary behaviors in daily living. Food cultures, for example, are deeply rooted in daily living and are characterized by traditional ethnic specific products and dishes.

“We are born with rice, no kidding, we grew up with rice, from the age of 6 months we eat rice”.

Several environmental factors were mentioned by individuals from all ethnic groups that might be relevant in optimizing protein intake. For example, factors related to food cultures were mentioned, characterized by the abundance of food during social activities, family visits and special festivities. In particular, Moroccan and Turkish older adults indicated that the social environment influences dietary habits, as they often eat together with their families and show great hospitality to guests.

“When people come to visit, we spend half a day in the kitchen to cook plenty of food for the guests, because we know that they expect that from us”.

Participants emphasize that the social cohesion and traditions contribute to strong dietary habits, which lead to the maintenance of their current dietary behaviors. Several Moroccan and Turkish participants reported not feeling comfortable enough to alter their diet in favor of protein without the support of their social environment. Surinamese older adults mainly named social activities and frequent festivities to be most challenging in changing dietary behaviors.

## 4. Discussion

To our knowledge this study was the first to examine the dietary protein intake and the behavioral and environmental factors that affect protein intake among older adults from ethnic minorities in the Netherlands, in order to look at how protein intake may be improved.

First, the amount of protein is important to consider in order to maintain muscle mass while aging [10]. Based on the recommended daily allowance of 0.8 g/kg BW/day cut-off, the majority of the ethnic populations from our study consume an adequate amount of protein. Still, 5–25% of the older adults from the different ethnic minorities, with exception of the Turkish, may improve their total daily protein intake to reach the RDA levels. Ethnic minority groups show higher sarcopenia prevalence rates compared to the Dutch majority populations [9], so they generally have a worse disease risk profile. Indeed chronic diseases and physical limitations appear at a younger age in Dutch ethnic minority groups [3,4,5]. Therefore 1.0–1.2 g of protein per kilogram bodyweight per day is suggested to be relevant for these populations to maintain or improve muscle mass [17,30]. Since 40% to 60% of older adults from Dutch ethnic minorities, except for Turkish men, do not reach the ESPEN recommendation, the amount of dietary protein should be improved to at least 1.0 g/kg BW/day.

Our quantitative analyses also pointed out that the largest proportion of the dietary protein intake in all ethnic groups originate from animal sources, and a smaller proportion originate from plant sources. This is comparable to other older populations in the Netherlands [31]. These results are beneficial from a physiological perspective, since animal protein, especially those from dairy, seems to be most beneficial to improve muscle protein synthesis compared to plant protein, and is, therefore, to be favored [20,21,32]. Unfortunately from this study, it is still unclear what the exact distribution is of protein sources throughout the day in these minority populations. More research is needed in this field.

From our qualitative analyses, we found indications that the timing of protein intake is suboptimal, due to the irregularity of daily schedules, thus, unconcise timing of daily meals. These findings correspond to recent quantitative findings from HELIUS showing that ethnic minority populations skip breakfast more often and eat more erratically [33]. Additionally, in an earlier qualitative study, Turkish and Moroccan participants in the Netherlands experienced a contrast between lifestyles in the Netherlands and lifestyles in their country of origin, inducing unconcise timing of meals [24]. Irregular eating patterns are specifically not favorable for maintaining muscle mass in older adults. Several studies show an importance of dietary protein intake distributed throughout the day to maintain muscle mass [34,35]. An amount of 25–30 g of protein per meal is needed to maximally stimulate skeletal muscle protein synthesis, where 20 g of high quality protein or 10 g of essential amino acids is considered the minimum [21]. Several studies suggest that a protein intake pattern that is evenly distributed throughout the day is most favorable to stimulate muscle protein synthesis in older adults [36,37]. Indeed, RCTs showed the effectiveness of a concise timing of protein throughout the day for muscle mass and physical functioning [15,16]. Therefore, a concise timing of the breakfast and other meals throughout the day should be advised to maintain and/or gain muscle mass in older adults from ethnic minorities.

Our qualitative study also provided information on personal and environmental factors that affect protein intake in these minority populations. The majority of the participants in the focus group discussions lack knowledge and awareness on protein containing food products and its importance for the aging process. They also indicated not to know how to consume adequate protein in daily living. Future interventions to optimize protein intake in these minority populations should include methods that target personal determinants as knowledge, skills, and awareness, to be able to improve dietary protein intake [38]. Additionally, our qualitative study showed that cultural background determines food choices, certain beliefs about health behaviors, and strong eating habits. This is in line with our quantitative analysis that showed that food products that contributed the most to daily protein intake were ethnic specific and varied between ethnic groups. Cultural background also seems to determine the influence of the social environment on dietary intake. These cultural and social influences on food consumption are also comprehensively discussed in earlier qualitative studies in these ethnic groups in the Netherlands [23,24]. The strong social norms, religion and cultural beliefs play a crucial role in dietary behaviors in ethnic minority groups [25]. Such determinants are deeply rooted and challenging to modify, so are expected not to be sustainable in changing behaviors. However, previous studies have shown the importance of cultural adaptations for effective culturally targeted interventions. As such, to account for specific cultural needs, a strategy to optimize protein intake in older ethnic minorities should consider, for example, ethnic specific dietary needs and active involvement of the social environment to increase social support and acceptance [39].

Based on the current study we advise to improve the amount of protein and timing of protein to be the windows of opportunity for optimizing protein intake in older adults from ethnic minorities. Hence, when aiming to improve muscle strength and physical functioning, the effectiveness of an intervention focusing on protein alone may not be enough. Several studies, including a meta-analyses, indicate that a higher amount of protein is not associated with increased muscle mass [40,41,42] or muscle strength [43]. A more effective strategy to improve muscle mass and physical functioning in these older ethnic minority populations should include a physical exercise component, preferably resistance exercise. Several trials showed the effectiveness of combined exercise and protein interventions in older adults, with protein intake levels well above the RDA [15,44,45,46]. As such, if combined with (resistance) exercise, increasing the amount of dietary protein to a minimum of 1.0–1.2 g/kg BW/day, might be a beneficial strategy to improve muscle mass and functioning in these populations. Based on a recent study among the HELIUS study population, which shows a lower percentage of the older adults from ethnic minorities meet the Dutch physical activity norm compared to the Dutch majority population, (resistance) exercise behavior should also be targeted [9].

## 5. Conclusions

In conclusion, the amount and timing of protein intake seem to be major concerns in older adults from ethnic minorities in the Netherlands. Optimizing protein intake in these groups requires a culturally sensitive approach, which accounts for ethnic specific protein product types and strong social norms. Future studies should therefore focus on developing and evaluating a culture-sensitive intervention that is community based to increase social support, and includes methods to improve awareness and skills related to protein intake among different ethnic minority populations. An intervention that manages to optimize protein intake may eventually contribute to improved muscle mass, strength, physical functioning, and quality of life in older adults from ethnic minorities in the Netherlands.

## Figures and Tables

**Table 1 nutrients-13-00184-t001:** Characteristics of the subsample from participants of Healthy Life in an Urban Setting (HELIUS) (N = 1415).

	Dutch	South Asian Surinamese		African Surinamese		Turkish		Moroccan	
**MEN**	**mean ± SD,** **N (%)**	**mean ± SD,** **N (%)**		**mean ± SD,** **N (%)**		**mean ± SD,** **N (%)**		**mean ± SD,** **N (%)**	
**N (number of cases)**	272 (19.2%)	140 (9.9%)		143 (10.1%)		32 (2.2%)		51 (3.6%)	
**Age (years)**	61.7 ± 4.0	60.7 ± 4.0		60.6 ± 4.2		58.5 ± 3.2		60.5 ± 4.2	
**BMI (kg/m^2^)**	26.3 ± 3.7	26.4 ± 3.7		26.6 ± 4.0		27.3 ± 3.0		28.2 ± 4.1	
**Energy intake (Kcal/day)**	2333 ± 587	2130 ± 751		2250 ± 739		2331 ± 648		2244 ± 843	
**Protein intake (g/day)**	88.9 ± 22.7	87.8 ± 34.5		89.2 ± 32.3		105.3 ± 28.7		95.5 ± 39.2	
**Protein intake (g/kg BW/d) ***	1.06 ± 0.31	1.20 ± 0.54	*p* = 0.024	1.13 ± 0.46	*p* = 0.613	1.38 ± 0.38	***p* = 0.001**	1.18 ± 0.53	*p* = 0.351
**Animal protein intake (g/kg BW/d) ***	0.63 ± 0.22	0.70 ± 0.41	*p* = 0.203	0.67 ± 0.36	*p* = 0.714	0.86 ± 0.36	***p* = 0.001**	0.65 ± 0.34	*p* = 0.993
**Plant protein intake (g/kg BW/d) ***	0.44 ± 0.17	0.50 ± 0.23	***p* = 0.024**	0.46 ± 0.19	*p* = 0.841	0.52 ± 0.16	*p* = 0.128	0.53 ± 0.28	***p* = 0.010**
**Protein intake (% EN)**	15.4 ± 2.6	16.4 ± 3.3		16.1 ± 3.4		18.1 ± 3.6		17.1 ± 3.4	
**>0.8 g/kg BW/d (%) ^$^**	226 (83.1%)	106 (75.7%)		113 (79.0%)		31 (96.9%)		36 (70.6%)	
**>1.0 g/kg BW/d (%) ^§^**	154 (56.6%)	78 (55.7%)		87 (60.8%)		29 (90.6%)		28 (54.9%)	
**WOMEN**	**mean ± SD,** **N (%)**	**mean ± SD,** **N (%)**		**mean ± SD,** **N (%)**		**mean ± SD,** **N (%)**		**mean ± SD,** **N (%)**	
**N (number of cases)**	290 (20.5%)	179 (12.7%)		230 (16.3%)		29 (2.0%)		49 (3.4%)	
**Age (years)**	61.6 ± 4.1	60.2 ± 4.1		59.7 ± 3.8		59.3 ± 4.0		59.6 ± 4.4	
**BMI (kg/m^2^)**	25.9 ± 4.5	28.0 ± 4.6		30.2 ± 5.9		32.9 ± 6.1		31.0 ± 5.2	
**Energy intake (Kcal/day)**	1888 ± 493	1752 ± 558		1748 ± 634		2122 ± 699		1783 ± 679	
**Protein intake (g/day)**	72.4 ± 21.1	74.3 ± 27.8		73.6 ± 28.1		91.3 ± 35.8		77.3 ± 32.0	
**Protein intake (g/kg BW/d) ***	1.04 ± 0.33	1.11 ± 0.44	*p* = 0.249	0.96 ± 0.42	*p* = 0.261	1.16 ± 0.48	*p* = 0.490	1.05 ± 0.52	*p* = 0.999
**Animal protein intake (g/kg BW/d) ***	0.61 ± 0.24	0.62 ± 0.34	*p* = 0.999	0.59 ± 0.32	*p* = 0.888	0.68 ± 0.35	*p* = 0.799	0.56 ± 0.36	*p* = 0.819
**Plant protein intake (g/kg BW/d) ***	0.42 ± 0.15	0.50 ± 0.20	***p* = 0.001**	0.38 ± 0.18	***p* = 0.033**	0.49 ± 0.22	*p* = 0.439	0.49 ± 0.24	*p* = 0.153
**Protein intake (% EN)**	15.4 ± 2.7	17.0 ± 4.0		17.1 ± 3.8		17.2 ± 3.3		17.2 ± 2.7	
**>0.8 g/kg BW/d (%) ^$^**	222 (76.7%)	131 (73.2%)		135 (58.7%)		23 (79.3%)		35 (71.4%)	
**>1.0 g/kg BW/d (%) ^§^**	143 (49.3%)	103 (57.5%)		96 (41.7%)		17 (58.6%)		24 (49.0%)	

BMI = body mass index, %EN = percentage of energy intake, g/kg BW/d = grams per kilogram bodyweight per day. * *p*-values represent comparison to Dutch majority population. ^$^ RDA (recommended daily allowance) of protein set by WHO (World Health Organization). ^§^ Minimum protein recommendation set by European Society for Clinical Nutrition and Metabolism (ESPEN) discussion group. *p* < 0.05 is marked as bold.

**Table 2 nutrients-13-00184-t002:** Subject characteristics of the focus group participants.

	South Asian Surinamese	African Surinamese	Turkish	Moroccan
	Mean, N	Mean, N	Mean, N	Mean, N
**N (number of focus group discussions)**	2	2	5	2
**N (number of participants)**	14	11	29	15
**Sex (men/women)**	2/12	0/11	9/20	12/3
**Age (years)**	74	80	64	-*

* The age of the Moroccan participants is unknown due to unwillingness to provide any personal details.

## Data Availability

Any researcher can request the data by submitting a proposal to the HELIUS Executive Board as outlined at http://www.heliusstudy.nl/en/researchers/collaboration, by email: heliuscoordinator@amsterdamumc.nl. The HELIUS Executive Board will check proposals for compatibility with the general objectives, ethical approvals and informed consent forms of the HELIUS study. There are no other restrictions to obtaining the data and all data requests will be processed in the same manner. The qualitative data presented in this study are available on request from the corresponding author.

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
