# Peer review of "Dietary Protein Intake in Older Adults from Ethnic Minorities in the Netherlands, a Mixed Methods Approach"

_nutrients, 2021, doi:10.3390/nu13010184_

Round 1

Reviewer 1 Report

Thank you for giving me an opportunity to review this manuscript. I have included below some comments.

Comment 1:

The purpose of this study is to investigate protein intake and the underlying behavioral and environmental factors that affect protein intake.

Although the author explains qualitative analyses, but I could not understand coded factors.

I would like you to elaborate the coded factors.

If there are factors coded by quantitative analysis, how did you examine the relationship between the factors encoded by the qualitative analysis and protein intake?

Author Response

Reviewer 1

Comment:

Thank you for giving me an opportunity to review this manuscript. I have included below some comments.

The purpose of this study is to investigate protein intake and the underlying behavioral and environmental factors that affect protein intake.

Although the author explains qualitative analyses, but I could not understand coded factors.

I would like you to elaborate the coded factors.

If there are factors coded by quantitative analysis, how did you examine the relationship between the factors encoded by the qualitative analysis and protein intake?

Response:

Thank you for your comment. We understand the unclarity of the exact coded factors in our qualitative analysis. To clearify this, the following sentence is added to the method section to make clear we explored specific (personal and environmental) behavioral determinants of protein intake (line 136-143):

‘The qualitative study focused on behavioral determinants affecting protein intake, both personal and environmental. First we explored personal determinants, such as  current knowledge and awareness about protein and its role in healthy ageing. Second we explored, environmental determinants that explain dietary (protein) behavior and habits’

We also made a clearer disctinction of these behavioral determinants by rearranging sentences throughout the results section of our qualitative analysis (please see lines 214-270)

To make clear the qualitative research is complementary to the quantitative research we altered the following sentence of our objective at the end of the introduction, emphasizing how the quantitative research relates to the qualitative research (line 74-75).

‘We carried out a mixed methods study to examine quantitively the dietary protein intake, including protein sources, and explore qualitatively its underlying personal and environmental behavioral determinants among older adults from ethnic minority groups in the Netherlands.’

Reviewer 2 Report

The paper is somewhat original; it is particularly suitable for this journal.
Substantial changes are not necessary.

The paper is interesting because it focuses its attention on ethnic minority populations, which are a substantial part of the society, almost never included in scientific studies, but frequent users of health care system.

The idea of studying minorities is conceptually important and original. It is interesting to note that awareness about the importance of protein intake on muscle health is lacking.

It would be appreciated if authors could give more space in the discussion on the specific weight of the religious-cultural factors vs lifestyle linked to social and job-related aspects, the latter being modifiable.

Practical examples of structured awareness campaigns would enrich “conclusions” section.

Author Response

Reviewer 2

Comment:

The paper is somewhat original; it is particularly suitable for this journal.
Substantial changes are not necessary.

The paper is interesting because it focuses its attention on ethnic minority populations, which are a substantial part of the society, almost never included in scientific studies, but frequent users of health care system.

The idea of studying minorities is conceptually important and original. It is interesting to note that awareness about the importance of protein intake on muscle health is lacking.

It would be appreciated if authors could give more space in the discussion on the specific weight of the religious-cultural factors vs lifestyle linked to social and job-related aspects, the latter being modifiable.

Practical examples of structured awareness campaigns would enrich “conclusions” section.

Response:

Thank you very much for your positive feedback on our work . Considering the comment on the additional factors in the discussion, we altered the following sentences (line 323-329):

‘The strong social norms, religion and cultural beliefs play a crucial role in dietary behaviors in ethnic minority groups [25]. Such determinants are deeply rooted and challenging to modify, so are expected not to be sustainable in changing behaviors. However, previous studies have shown the importance of cultural adaptations for effective culturally targeted interventions. As such, to account for specific cultural needs, a strategy to optimize protein intake in older ethnic minorities should consider for example ethnic specific dietary needs and active involvement of the social environment to increase social support and acceptance [39].’

We also added more practical implications from our study in the conclusion (line 351-355):

‘Future studies should therefore focus on developing and evaluating a culture-sensitive intervention that is community based to increase social support and includes methods to improve awareness and skills related to protein intake among different ethnic minority populations.’

Reviewer 3 Report

The manuscript covers a current theme and brings along insights into relatively less-known world of ethnic minorities in Western Europe. The research design, incorporating both a quantitative and a qualitative part, matches the goal very well, as we often are not able to understand the mechanisms and motivations behind the observed quantitative values. The inclusion of a chemical analysis of ethnic food can be viewed as an additional  strength of the study.

As for the study group, it includes, despite the title of the manuscript, mostly relatively young subjects with mean age between 58.5 and 61.7 years. "Older adults" are usually classified as having at least 65 (UN criteria) or 60 (WHO criteria) years of age. It would thus be way more interesting to perform the study on a 65+ (or 60+) age group. This is the more important as persons aged 55 are way less likely to have problems with their muscle mass and strength than those aged 65 or 75, in whom sarcopenia is more prevalent and translates, practically, into worse (less independent) functioning. As it is probably not realistic to demand a re-study, it is advisable to change the title to include the term "55+" instead of "older adults".

Introduction lines 55-56: it would be appropriate to indicate that exceeding the daily intake of 1.2 gram of protein per kilogram bodyweight per day in older persons should - as commonly accepted in geriatrics - trigger caution regarding the renal function; for higher daily allowances medical supervision is suggested.

The conclusions could indicate potential ways of intervention - to fully exploit the results.

Regarding the spelling, please stick consistently to the US one.

Author Response

Comment:

The manuscript covers a current theme and brings along insights into relatively less-known world of ethnic minorities in Western Europe. The research design, incorporating both a quantitative and a qualitative part, matches the goal very well, as we often are not able to understand the mechanisms and motivations behind the observed quantitative values. The inclusion of a chemical analysis of ethnic food can be viewed as an additional  strength of the study.

As for the study group, it includes, despite the title of the manuscript, mostly relatively young subjects with mean age between 58.5 and 61.7 years. "Older adults" are usually classified as having at least 65 (UN criteria) or 60 (WHO criteria) years of age. It would thus be way more interesting to perform the study on a 65+ (or 60+) age group. This is the more important as persons aged 55 are way less likely to have problems with their muscle mass and strength than those aged 65 or 75, in whom sarcopenia is more prevalent and translates, practically, into worse (less independent) functioning. As it is probably not realistic to demand a re-study, it is advisable to change the title to include the term "55+" instead of "older adults".

Response:

We appreciate your comment on using the term 'older adults' for a group aged >55 years. As being mentioned in the paper (line 36-45), ethnic minority populations may suffer from metabolic diseases and sarcopenia ten years earlier compared to our Dutch population, which has been published earlier by our group (Dorhout, 2020) Therefore, we use the term 'older adults'  with age 55 as cutoff in these populations. To clarify this decision, we altered the following sentence in the introduction (line 40):

‘For instance, in the Netherlands, chronic diseases and physical limitations appear on average ten years earlier compared to the Dutch majority population [3-7]’

Comment:

Introduction lines 55-56: it would be appropriate to indicate that exceeding the daily intake of 1.2 gram of protein per kilogram bodyweight per day in older persons should - as commonly accepted in geriatrics - trigger caution regarding the renal function; for higher daily allowances medical supervision is suggested.

Response:

Thank you for the suggestion to add information about the risk of high protein intake for renal function. We believe, however, that this is not really needed. There is ample data that suggests that higher protein intakes would not harm renal function in those that do not suffer from renal insufficiency (eGFR <30ml/min) (PMID: 30383278). In addition, current intakes of the populations studied in our study are not even close to upper limits. As this information would distract from our main message we have decided not to add a comment. 

Comment

The conclusions could indicate potential ways of intervention - to fully exploit the results.

Response:

Thank you for the suggestion to clarify some practical implications for future interventions on optimizing protein intake in ethnic minority populations. Accordingly, we added the following line in the conclusion section (line 351-355):

‘Future studies should therefore focus on developing and evaluating a culture-sensitive intervention that is community based to increase social support and includes methods to improve awareness and skills related to protein intake among different ethnic minority populations.’

Comment:

Regarding the spelling, please stick consistently to the US one.

Response:

We have made changes accordingly.

Round 2

Reviewer 1 Report

Thank you for answer all my questions.